# Isolation and Characterization of a New Endophytic Actinobacterium *Streptomyces californicus* Strain ADR1 as a Promising Source of Anti-Bacterial, Anti-Biofilm and Antioxidant Metabolites

**DOI:** 10.3390/microorganisms8060929

**Published:** 2020-06-19

**Authors:** Radha Singh, Ashok K. Dubey

**Affiliations:** Department of Biological Sciences & Engineering, Netaji Subhas University of Technology, New Delhi 110078, India; dharana.radha@gmail.com

**Keywords:** endophytic actinobacteria, *Streptomyces* sp., anti-*S. aureus*, anti-MRSA, anti-biofilm

## Abstract

In view of the fast depleting armamentarium of drugs against significant pathogens, like methicillin-resistant *Staphylococcus aureus* (MRSA) and others due to rapidly emerging drug-resistance, the discovery and development of new drugs need urgent action. In this endeavor, a new strain of endophytic actinobacterium was isolated from the plant *Datura metel*, which produced secondary metabolites with potent anti-infective activities. The isolate was identified as *Streptomyces californicus* strain ADR1 based on 16S rRNA gene sequence analysis. Metabolites produced by the isolate had been investigated for their antibacterial attributes against important pathogens: *S. aureus*, MRSA, *S. epidermis*, *Enterococcus faecium* and *E. faecalis*. Minimum inhibitory concentration (MIC_90_) values against these pathogens varied from 0.23 ± 0.01 to 5.68 ± 0.20 μg/mL. The metabolites inhibited biofilm formation by the strains of *S. aureus* and MRSA (Biofilm inhibitory concentration [BIC_90_] values: 0.74 ± 0.08–4.92 ± 0.49 μg/mL). The BIC_90_ values increased in the case of pre-formed biofilms. Additionally, the metabolites possessed good antioxidant properties, with an inhibitory concentration (IC_90_) value of 217.24 ± 6.77 µg/mL for 1, 1-diphenyl-2-picrylhydrazyl (DPPH) free radical scavenging. An insight into different classes of compounds produced by the strain ADR1 was obtained by chemical profiling and GC-MS analysis, wherein several therapeutic classes, for example, alkaloids, phenolics, terpenes, terpenoids and glycosides, were discovered.

## 1. Introduction

The emergence of drug resistance among pathogens has assumed alarming proportions in recent times, causing rapid depletion in the current armamentarium of drugs to fight such infections [1]. This has posed a serious threat to human health globally and required the rapid development of new and effective drugs at a pace faster than resistance to achieve desirable outcomes in the treatment of infectious diseases. Some of these pathogens have gained exceptional notoriety due to their tremendous ability to adapt, to evade the host immune response and to develop drug resistance. This has led World Health Organization to enlist significant human pathogens under critical, high and medium priority categories [2]. *Enterococcus faecium* (vancomycin resistant) and *Staphylococcus aureus* (methicillin resistant) are considered as high priority pathogens for whom new antibiotics are required the most urgently. Infections involving such pathogens are often associated with biofilms, which are responsible for multi-fold increase in drug-resistance of the pathogens [3,4]. Therefore, it is highly desirable that the new antibiotics possess anti-biofilm activities: disruption of pre-formed biofilms and inhibition of biofilms formation. 

Some of the recent studies have reported that the reactive oxygen species were causing antibiotic tolerance in *S. aureus* during systemic infections [5]. Further, oxidative immune response of the host appeared to be switched on during bacterial infections, resulting in increased oxidative stress to the host [6]. Additionally, the antibiotics used to treat infections might also cause an increase in the level of oxidative stress [7,8]. It was reported that antioxidants might prevent oxidative stress-induced pathology [9]. Biofilm formation in *S. aureus* is also enhanced in the presence of oxidative stress [10]. Therefore, providing antioxidants may help in the inhibition of biofilms formation and thus in the prevention of concomitant resistance to antibiotics among the pathogens. 

In view of the facts mentioned above, our research endeavors have focused on the discovery of novel anti-infective therapeutics for the treatment and cure of drug resistant infections. Characterization of antioxidant properties of metabolites was also part of our study design due to its foreseeable application in therapy of infectious diseases. There are several approaches to develop drugs, for example, rational drug design (structure-based design of inhibitors against target), synthetic and combinatorial chemistry, high throughput screen of chemical libraries and mining of natural products [11,12]. However, from the stand point of discovery of novel pharmacophore or new classes of drug working on as yet unknown targets, mining of natural products is the obvious choice. Therefore, we chose natural products in the search for new antibiotics. Furthermore, we have considered actinobacteria from vast pool of natural product resources due to their versatility, ubiquity and ability to produce therapeutic compounds with extensive chemical diversity [13,14]. The genus *Streptomyces* of actinobacteria has been regarded as containing the most prolific producers of therapeutic compounds [15,16]. However, repeat discovery of known molecules remains a challenge while hunting the actinobacteria for drugs [17,18]. One of the possible approaches to avoid the repeat discovery of the drugs could be the sourcing of actinobacteria from niche habitats instead of common sources like soil. Accordingly, we have explored endophytic actinobacteria in an effort to enhance the chances of finding new compounds as potential drug candidates. Endophytic actinobacteria are the microorganisms that reside within the plant tissues without causing any adverse effect to plants [19]. Further, for increasing the prospect of strain novelty, we have selected a medicinally important plant, *Datura metel*, which had largely remained unexplored for endophytic population of actinobacteria. 

In the present communication, identification of a novel strain, *Streptomyces californicus* strain ADR1 is reported from the plant *D. metel*. Secondary metabolites produced by the isolate ADR1 were characterized for their antibacterial, antibiofilm and antioxidant properties. Further, the metabolite preparations were analyzed for different class of therapeutically significant compounds produced by the isolate.

## 2. Materials and Methods 

### 2.1. Isolation of Endophytic Actinobacteria

The endophytic actinobacteria were isolated from the plant, *Datura metel*. The ex-plants were surface sterilized by following the method reported elsewhere [20]. The sterilized plant parts were aseptically grinded by using autoclaved mortar pestle in phosphate-buffered saline (PBS) pH 7.0. The ground paste was spread over the following isolation media: nutrient agar, asparagine glycerol (AGS) agar) [21], humic acid—vitamin agar [22] and starch casein nitrate (SCN) agar [23]. The media were supplemented with cycloheximide (50 µg/mL). The plates were incubated at 28 °C for up to four weeks with regular observations for potential actinobacterial colonies. 

The putative actinobacterial colonies were transferred and maintained on AGS medium. Purification of the isolates was achieved by repeated cycles of streaking on fresh plate. The purified cultures were screened for anti-bacterial against *S. aureus* ATCC 29213 by well diffusion method [24]. 

### 2.2. Molecular Identification and Characterization of the Isolate ADR1

Molecular identification of the strain ADR1 was based on 16S rRNA gene sequence analysis. The genomic DNA of the strain ADR1 was isolated using the method developed for Gram-positive bacteria [20] with a few modifications. Briefly, amplification of 16S rRNA gene was carried out using universal primers: V1f (50-AGAGTTTGATCMTGGCTCAG-30), V9r (50-AAGGAGGTGATCCANCCRCA-30), V3f (50-CCAGACTCCTACGGGAGGCAG-30) and V6r (50-ACGAGCTGACGACARCCATG-30) in a PCR machine (Mastercycler^®^ nexus, Eppendorf International, Germany) by using the programme described elsewhere [25]. The amplified product was sequenced by Sanger’s method using a 3130XL sequencer (Applied Biosystems, California, USA) for the 16S rRNA gene using universal primers as described above. The sequences were aligned in MEGA 6.0 to generate single consensus sequence. Homology search was performed using the standard Basic Local Alignment Search Tool (BLAST) sequence similarity search tool of the NCBI database to establish the identity of the isolate ADR1. Nucleotide sequences producing significant alignments after BLAST analysis with the 16S rRNA gene sequence of ADR1 were retrieved in FASTA format. These sequences were used to generate phylogenetic relationship of ADR1 with them by using the software, Phylogeny.fr [26,27]. The analysis was done by using advanced mode of this tool, which is an automated programme that performs step-by-step analysis starting from the multiple alignment of the sequences (MUSCLE 3.8.31) [28], alignment curation (Gblocks 0.91b), construction of phylogenetic tree (PhyML 3.1/3.0 aLRT) [29,30] to the visualization of phylogenetic tree (TreeDyn 198.3) [31]. The culture was characterized for its morphological features on different international *Streptomyces* protocol (ISP) media [32]. Isolate ADR1 was streaked on ISP1, ISP2, ISP3, ISP4, ISP5, ISP6 and ISP7 media plates and was incubated for 7 days at 28 °C for phenotypic and morphological observations. Single colony morphology of the culture was observed under Nikon stereo zoom microscope SMZ1270 at zooming ratio of 12.7:1 and resolution of 8×. Mycelial structure was observed under Nikon E600 microscope (Nikon, Tokyo, Japan) at a resolution of 100×. 

### 2.3. Production of Secondary Metabolites

A single colony from freshly grown culture plate (72 h) was inoculated in 50 mL SCN broth (pH 7.4), which was incubated at 28 °C for 72 h to develop the pre-seed culture. The production medium (SCN broth, pH 7.2) was inoculated with the pre-seed culture (1%; *v*/*v*) to commence production of the secondary metabolites, which was carried out for 7 days at 28 °C in an incubator shaker (Adolf Kuhner AG, Birsfelden, Basel, Switzerland) run at 200 rpm. The cell-free broth was recovered by centrifugation at 5000× *g* for 20 min in Sorvall RC 5C plus centrifuge (Kendro Laboratory Products, Newtown, Connecticut, USA). The metabolites were recovered from the supernatant by using liquid-liquid extraction with equal volume of ethyl acetate. The extracted metabolites were dried by using rotary evaporator (50 °C) and vacuum oven (35 °C). The dried metabolite preparations were stored at 25 °C ± 2 °C till further use. 

### 2.4. Antimicrobial Susceptibility Testing

The reference strains of bacterial pathogens used in this study were: *S. aureus* ATCC 29213, *S. aureus* ATCC 25923, *S. aureus* ATCC 13709, MRSA ATCC 43300, MRSA 562, *S. epidermis* ATCC 12228, *Enterococcus faecalis* ATCC 29212, *E. faecium* ATCC 49224 and *E. faecium* AIIMS. In-vitro antibacterial activity of the metabolite extract was determined on cation adjusted Muller Hinton agar (MHA) (Himedia, Mumbai, India) plates using well diffusion method [24]. The minimum inhibitory concentration (MIC_90_) values were measured in a 96-well microtiter plates by the broth microdilution method as per the guidelines of Clinical and Laboratory Standards Institute (CLSI) [33]. Briefly, a stock solution of the metabolite extract (1mg/mL) was prepared in 0.2% DMSO and cation adjusted Muller Hinton broth. Bacterial pathogens (100 µL; 2 × 10^8^ CFU/mL) and metabolite extract (100 µL) at concentrations varying from 125 to 0.122 µg/mL were added to the individual well in the microtitre plate. A sample control (ADR1 extract alone) and blank (media only) were included in each assay. After incubation for 24 h at 37 °C, iodonitrotetrazolium chloride (INT) (Sisco Research Laboratories, Mumbai, India) was added to the wells and the plates were incubated further for 30 min. The absorbance was measured on a multimode reader (Biotek Instruments, Winooski, Vermont, USA) at 490 nm. The value of MIC_90_ was considered to be the minimum concentration at which no visible growth could be observed. The following equation was used to compute the percent inhibition [34].
Growth inhibition of pathogen (%) = [(control OD_490 nm_ − test OD_490 nm_)/control OD_490 nm_] × 100(1)

### 2.5. Antibiofilm Assay 

Biofilms of *S. aureus* ATCC 25923, *S. aureus* ATCC 29213, MRSA ATCC 43300 and MRSA 562 were produced by using the method published elsewhere [35] in accordance with the CLSI guidelines [33]. Briefly, overnight grown reference cultures were suspended in tryptic soy broth (Himedia, Mumbai, India) supplemented with 2% glucose to attain turbidity equivalent to 0.5 McFarland standard (2 × 10^8^ CFU/mL). A total of 100 μL of the cell suspension was transferred to the wells on the microtiter plate and was incubated at 37 °C for 24 h under static condition. Non-adherent cells were aspired out along with the medium. The wells were rinsed with 100 μL of phosphate-buffered-saline (PBS). Fresh medium containing desired concentrations of ADR1 metabolites (from 250 to 0.49 µg/mL) were added to the wells on the microtiter plate, which was then incubated for the next 24 h at 37 °C under static condition. Viability of the biofilms was quantified by INT-calorimetric assay as described above. The following equation was used to find % inhibition of biofilm [35].
Biofilm inhibition (%) = [(control OD _490 nm_ − test OD _490 nm_)/control OD _490 nm_] × 100(2)

### 2.6. Antioxidant Activity

The antioxidant potential of ADR1 metabolites was assessed by measuring reduction of DPPH (1, 1-diphenyl-2-picrylhydrazyl) free radicals as reported earlier [36]. Briefly, DPPH solution (0.1 mM) was prepared in methanol; 100 µL of this solution was added to 100 µL of the ADR1 metabolite preparations at different concentrations varying from 1000 to 7.81 μg/mL in 96-well microtitre plate. The plate was then incubated at 25 °C for 20 min in dark and the absorbance was measured at 517 nm. The scavenging strength was calculated using the following formula [36].
% scavenging activity = [(absorbance of DPPH control − absorbance of DPPH in the presence of metabolite)/absorbance of DPPH control] × 100(3)

### 2.7. Hemolytic Activities

Hemolytic activity was determined by disc diffusion assay using sheep blood agar (SBA) plates (Himedia, Mumbai, India). A total of 10 µL solution containing varying concentrations of the ADR1 metabolites (1000 to 7.8125 μg/mL) were dispensed on discs placed aseptically on the SBA plates and were incubated for 24 h at 37 °C. The type of hemolysis was observed as alpha, beta and gamma [37].

### 2.8. Secondary Metabolite Profiling and GC-MS Analysis

The ADR1 metabolite extract was used at a concentration of 5 mg/mL for chemical profiling of the classes of metabolites present in the extract. The tests for the different class of metabolites, for example, anthraquinones, glycosides, terpenoids, flavonoids, tannins, alkaloids, saponins, sterols, anthocyanins, coumarins, tannins, lactones, terpenes, fatty acids, proteins/amino acids and carbohydrates were carried out by using standard methods reported earlier [38,39,40].

Further analysis of the metabolites was carried out by employing GC-MS (GC-MS-QP2010 plus; Shimadzu, Kyoto, Japan) as outlined below. A constant column flow rate of 1.21 mL/min with helium gas was maintained in RESTEK capillary column (30 m × 0.25 mm I.D. × 0.25 µm film thickness). Initial oven temperature was 100 °C for 3 min, which was increased to 250 °C for a hold time of 5 min, was further increased gradually to 280 °C where it was kept constant for 15 min. A total of 3 µL of sample (3 mg/mL) was injected in split mode (split ratio of 10.0) and linear velocity of the column was maintained at 40.9 cm/s. The mass fragmentation patterns (spectra) of the metabolites were obtained at electron ionization (EI) of 70 eV scanned over a m/z range of 40–650. The compounds detected were identified on the basis of comparison of the mass spectra with those available in the NIST14 and Wiley8 spectral library. The spectra having a match limit value lower than 700 were not considered.

### 2.9. Statistical Analysis 

All the experiments were performed in triplicates. The data were expressed as mean ± standard deviation. The statistical analysis and significance of the test was performed by analysis of variance (ANOVA), using the software, graph pad prism 5.01. The graphs were also generated using grouped analysis in the graph pad prism and represented with SEM in the form of error bar.

## 3. Results and Discussion

Eight putative actinobacterial endophytes, designated as ADR1 to ADR8, were isolated from the plant, *Datura metel*. While no isolate could be found from the stem part of the plant, six were obtained from the root tissues and two were from the leaves. The antibacterial potency of these isolates was examined against the reference strain, *S. aureus* ATCC 25923. Based on the size of the zone of inhibition, the isolate ADR1 was chosen for further studies as it produced largest zone (22.5 ± 0.58 mm). Production of the metabolites by ADR1 was carried out under the conditions as described under Section 2.3. The metabolite extract was recovered as red colour hygroscopic sticky mass (approximately 120 mg/L). 

### 3.1. Identification and Characterization of the Isolate ADR1

Amplification of the 16S rRNA gene from the genome of ADR1 produced a sequence of 1452 nucleotides. Blast analysis revealed 99.17% sequence identity of the ADR1 sequence with *S. californicus* strains with a query coverage of 99%. The phylogenetic relationship of the strain ADR1 can be seen in Figure 1, where it showed closest relationship with *S. californicus* strains. The 16S rRNA gene sequence obtained in this study was submitted as ‘*Streptomyces californicus* strain ADR1’ to NCBI GenBank with accession no. KU299789.1. 

A few strains of *S. californicus* had been reported earlier from the soil [41,42,43,44]. However, there are no reports of any endophytic strain of *S. californicus* till date to the best of our knowledge, making the present isolate as a new endophytic actinobacterial strain, designated as *S. californicus* strain ADR1. 

The strain ADR1 was characterized for its cultural attributes on ISP media 1 to 7. The results (Table 1) suggested that the extent of growth of the culture varied from scanty to abundant on different ISP media. Further, differences with respect to the colour of substrate and aerial mycelia, and production of diffusible pigments were also noted as described in the Table 1. When compared with the cultural characteristics of non-endophytic *S. californicus* strains JCM 6910, MNM-1400 and G16, it was observed that ADR1 shared a few similarities, for example, colour of aerial mycelium on ISP 2, 3 and diffusible pigments on 4, with *S. californicus* strain JCM 6910, a soil isolate from Japan [41,44,45]. However, the growth of ADR1 was abundant on ISP3, while that of JCM 6910 was poor. No diffusible pigment was produced by ADR1 in ISP5, while violet pigment was produced by JCM 6910. Other soil isolates of *S. californicus,* strain MNM-1400 and strain G16, were morphologically very different from the strain ADR1 [44,45] Thus, the endophytic *S. californicus* strain ADR1 was evidently distinct from the soil isolates.

A detailed view of the morphology was obtained through the study of single colonies (Figure 2; Panel A and B). Growth on different ISP media produced differences in colour and appearance of the colonies, which appeared as dense, depressed and rocky on ISP1, while on ISP2, 3 and 5 the aerial mycelia appeared to be fluffy and dusty. Clear exudates can be seen over the colony on ISP6. The colonies on ISP7 appeared scantly grown lacking distinct structures that were observed on other media. Prominent differences in the extent of growth, structure and pigmentation of the colonies on different ISP media were consistent with the earlier reports [46]. Such media-dependent phenotypic variations suggested that the primary and secondary metabolism of the culture varied significantly with changes in composition of the medium, which is in agreement with the current understanding of the physiology of the genus *Streptomyces* [47,48]. The microscopic observations showed highly branched flexuous mycelium and the arrangement of spores in a chain inside mycelium as shown in (Figure 2, panel C and D). These are some explicit features of *Streptomyces* species [32].

### 3.2. Antibacterial Spectrum of ADR1 Metabolites Against Significant Gram-Positive Pathogens

The spectrum of activity of the ADR1 metabolites against different Gram-positive pathogens was determined as described under Section 2. The results presented in Table 2 showed that the metabolite extract possessed broad spectrum activity against Gram-positive pathogens as it effectively inhibited growth of all the reference strains. It may be noted that the activities of metabolite extract against MRSA strains were similar to those against *S. aureus* strains. 

However, the above data provided more of a qualitative rather than quantitative estimate of the antibacterial activity. An accurate view of the potency of the metabolites was achieved by assessment of MIC_90_ values against the pathogenic strains used in this study. As presented in Figure 3a, the MIC_90_ values of the ADR1 metabolites against different strains of *S. aureus* were between 0.44 ± 0.07 − 0.84 ± 0.03 μg/mL, while against *S. epidermis* ATCC1222 it was even lower (0.23 ± 0.01 μg/mL). It may be noted that the potency against MRSA strains (Figure 3b) was as good as it was against *S. aureus* strains. These results can be considered as significant since the MIC_90_ values of the ADR1 metabolites stood better than the standard drug vancomycin (0.5 to 2 μg/mL) for MRSA strains [49]. Other than *S. aureus* strains, the MIC_90_ values of 1.92 ± 0.03 and 3.35 ± 0.18 μg/mL were observed against *E. faecium* strains AIIMS and ATCC 49214, respectively, which demonstrated good sensitivity of these strains to ADR1 metabolites. *E. faecalis* ATCC 29212 was the least sensitive to ADR1 metabolites among the reference pathogens since its MIC_90_ value (5.68 ± 0.20 μg/mL) stood at the maximum, but it was still better than some of the previously reported activities [50,51]. The MIC_90_ values of the metabolites from some of the recently reported *Streptomyces* spp. against the strains of *S. aureus* and MRSA were in the ranges of 2–125 μg/mL [52,53,54], which indicated wide variation in the anti-bacterial potency of the metabolites. The results of the present study clearly demonstrated that the anti-bacterial potency of ADR1 metabolites appeared at the bottom of the range or even lower, validating its promising potential for discovery of drugs against priority pathogens.

### 3.3. Anti-Biofilm Activity of ADR1 Metabolites Against S. aureus and MRSA

Biofilm-associated infections posed a greater challenge in treatment of infectious diseases as it is one of the major contributing factors in enhancing antibiotic-resistance among *S. aureus* and its methicillin-resistant strains [3]. Therefore, the discovery of drugs with potent anti-biofilm activity is needed more than ever before to combat the ever-growing global challenge of antibiotic resistance against the notorious pathogens like *S. aureus* and MRSA. In view of this, the biofilm inhibitory potential of the ADR1 metabolites was investigated. The results (Figure 4a) suggested that the ADR1 metabolites were able to effectively inhibit formation of biofilm by the *S. aureus* and the MRSA strains. Up to 90% reduction in the formation of biofilm could be achieved at significantly lower concentration of the metabolites; the BIC_90_ values were noted to be in the range of 0.74 ± 0.08 to 4.59 ± 0.71 μg/mL. The effectiveness of the ADR1 metabolites in the inhibition of the biofilm was found to be better than the previously reported activity of biofilm inhibition by actinobacterial metabolites where maximum inhibition of 83% was recorded at 265 μg/mL [35]. However, when the activity against pre-formed biofilms (24 h) was examined, the BIC_90_ values for *S. aureus* strains increased by many folds, for example, up to 45.69 ± 3.32 and 89.54 ± 0.40 μg/mL for *S. aureus* ATCC 25923 and *S. aureus* ATCC 29213, respectively. The biofilms produced by MRSA proved even more resistant (Figure 4b). It was reported that some of the well-known antibiotics like pyrrolomycin and related compounds could inhibit the biofilm only up to 67%–87% [55]. Similarly, in a study on the effect of antibiotics like rifampicin, polymyxin B, kanamycin and doxycyclin on reduction of *S. aureus* biofilm formation, only rifampicin was found to inhibit the biofilm by about 50% [56]. The effect of ADR1 metabolites on inhibition of biofilm formation as well as on the preformed biofilms was better than some previously reported metabolite extracts [57,58,59]. Inhibition of biofilm formation strongly suggested that the metabolites prevented adherence of *S. aureus* and MRSA cell to the polystyrene surface. Further, their ability to disrupt pre-formed biofilms might limit the biofilm-associated drug resistance among the pathogens.

### 3.4. Antioxidant Activity of the ADR1 Metabolites

Antioxidants assume significance for therapeutic applications in view of their role in neutralizing reactive oxygen species in patients fighting diseases involving infectious agents or metabolic disorders [60,61,62]. Hence, it was prudent to examine if the ADR1 metabolites possessed any such activity. The standard assay, which measured the reduction of DPPH free radical, revealed the antioxidant properties of ADR1 metabolites (Figure 5). The free radical scavenging activity of the metabolite extract and of ascorbic acid (a well-known antioxidant agent), followed a different pattern, where a sharp increase in DPPH scavenging activity (from 40% to 80%) was observed when concentration of the metabolite was increased from 62.5 to 125 µg/mL. However, in the case of ascorbic acid, there was no significant change in the activity over the above concentration range. The IC_90_ value for DPPH scavenging by ADR1 metabolite was achieved at the concentration of 217.24 ± 6.77 µg/mL, while that for the ascorbic acid it was 904.32 ± 12.93 µg/mL (Figure 5), which was approximately 4-fold higher compared to ADR1 metabolites. However, interestingly IC_50_ of the ascorbic acid was 4.617 ± 0.89 µg/mL while that of ADR1 extract was 77.41 ± 1.02 µg/mL (not plotted). It was recently reported that the endophytic actinobacterial strains BPSAC77, 101, 121 and 147 showed DPPH scavenging, with an IC_50_ value of 43.2 µg/mL, but the IC_90_ value was not reported in this study [63]. In another study, *Streptomyces* sp. had been found to scavenge DPPH free radicals at a much higher concentration, with an IC_50_ value at 435.31 ± 1.79 μg/mL [64]. In one of the recent studies, it was reported that IC_50_ values for DPPH scavenging activities of several isolates of actinobacteria varied from 12 ± 1.8 to 65 ± 3.2 μg/mL [65]. Such variations were also noted by others [54]. These reports indicated that there was a wide variation in the antioxidant activities of the metabolites produced by different actinobacterial strains. An in-depth characterization of relevant compounds in pure form may offer further insight into such differences in the activities.

### 3.5. Haemolytic Activity

Drug-induced haemolytic anaemia (DIHA) and thrombocytopenia (DIT) are common adverse effects associated with antibiotics [66]. Reports suggested that the haemolytic effects of antimicrobial peptides tyrocidine A and gramicidin S, limited their use as topical agents [67,68]. Ceftriaxone, a third-generation cephalosporin was also reported to cause haemolysis, attracting advise for restricted administration of the drug [69]. So, it is very important for a potential drug to be tested for its haemolytic activity to secure critical information on its clinical suitability. In this context, the haemolytic effect of the ADR1 metabolites was tested. It was noted that the ADR1 metabolites did not show any sign of haemolysis in the concentration range from 7.8125 to 1000 µg/mL (Figure 6), which was far greater than the MIC_90_ values against test pathogens. 

In some previous studies, the extracts with low or no haemolytic activity were considered suitable for further characterization of their antimicrobial properties [70]. This implied that the metabolite extract of strain ADR1 could be safe for further investigation into specific metabolites as probable drug candidates.

### 3.6. Secondary Metabolite Profiling and GC-MS Analysis of Metabolite Extract

Characterization of bioactivities of the metabolites had demonstrated that the present endophytic strain, *S. californicus* ADR1 was an unexplored source of antimicrobial, antibiofilm and antioxidant agents, which might have potential clinical applications. It was therefore prudent to analyze the composition of the metabolite extract to unravel the class of compounds produced. Therefore, the ADR1 metabolite extract was subjected to chemical profiling using specific reagents. The tests revealed the classes of compounds present in the extract of metabolites produced by the strain ADR1, which included anthraquinones, anthocyanins, terpenoids, terpenes, flavonoids, phenols, alkaloids and glycosides (Table 3). Various compounds, which belonged to these classes have therapeutic significance and are routinely reported from phytochemical screening of plant extracts. But recent literature suggested that actinobacteria too are proving to be excellent sources of such class of compounds [14,71]. 

While the results of chemical profiling were primarily qualitative in nature suggesting the presence or the absence of a specific chemical class, it did not offer any clue about the identity of the compounds. But it is important to have a deeper insight into the chemical class and identity of the specific compounds produced by the culture if these are to be considered for drug development. In pursuit of this goal, GC-MS analysis of the ADR1 metabolites was undertaken to probe the identity of metabolites based on their mass spectrum. 

The GC-MS analysis revealed the presence of alkaloids, terpene, terpenoids and glycosides in the ADR1 metabolite extract, thus confirming the findings of chemical profiling. The compounds were assigned probable identities based on similarity index (SI) value, generated from the mass spectral analyses of the test compounds and their comparison with the reference compounds listed in the NIST and Wiley library. An SI of more than 85 was considered for assignment of probable identity to the compounds [72]. Out of the total 30 peaks detected in the chromatogram, 20 peaks were specific to the culture *S. californicus* ADR1 while rest of the peaks were related to the production media and the solvent control (Appendix A). Only six of ADR1 metabolite peaks were found to have SI value greater than 85. Thus, identities could be assigned to only these compounds (Table 4). Majority of the compounds detected in the GC-MS chromatogram showed lower degree of similarity with SI values between 60–80 (Appendix A), suggesting a high probability of occurrence of novel compounds among the secondary metabolites produced by *S. californicus* strain ADR1.

The first compound, detected at RT 11.945 min, showed SI of 88 with Methanoazulen-9-ol, decahydro-2, 2, 4, 8-tetramethyl-stereoisomer, which belonged to the class sesquiterpene and possessed antibacterial as well as antioxidant properties [73]. This compound was earlier reported as a major component of plant essential oils [74,75] but there are no reports on this from microbial sources as yet. Another compound in ADR1 metabolite extract was detected at RT 14.519 min, which showed SI of 90 with 5-z-methyl-2-z-hydroxycarbonyl-5-e-ethenyl-4-z-propen-2-ylcyclohexanone, was a terpenoid having structure similar to Asperaculane B, a known GABA-transaminase inhibitor having significance in epilepsy treatment [76]. Terpenes and terpenoids have wide therapeutic potential like antimalarial, antibacterial, antiinflammatory, antioxidant as well as wound healing properties [77]. A flavonoid that was found most abundant among the ADR1 metabolites, showed SI of 85 with 4′-Methoxy-2′-(trimethylsiloxy) acetophenone. Flavonoids are well regarded molecules with broad-spectrum biological activities as well as applications in nutraceuticals and cosmetic industry [78]. However, there are no reports on the activity of this molecule till date. An alkaloid with SI of 88 in comparison to pyridineethanamine, n-methyl-n-[2-(4-pyridinyl) ethyl] was detected among the ADR1 metabolites at RT 16.068 min. The compound is known for its importance in the treatment of vertigo [79]. Alkaloids are well regarded for their wide spectrum of therapeutic activities including antibacterial, antioxidant, anti-inflammatory, anti-diabetic, antimalarial and anticancer properties [80]. The presence of therapeutically important chemical classes of compounds in ADR1 metabolite extract underscored its importance in the discovery of novel compounds with prominent therapeutic potential. 

## 4. Conclusions

The metabolites produced by *S. californicus* ADR1 had shown very low MIC_90_ values against a range of Gram-positive pathogens including those notified by the WHO as high priority pathogens. This suggested that it could be an excellent source for effective anti-infective compounds against some of the most challenging pathogens like *S. aureus*, MRSA, *S. epidermis, E. faecalis* and *E. faecium*. The ADR1 metabolites came across as potent inhibitors of biofilms of both methicillin-sensitive and methicillin-resistant strains of *S. aureus*. Along with the antimicrobial activity, the metabolites exhibited good antioxidant activity. Antioxidant property with no haemolytic activity indicated the potential of ADR1 metabolites for use as antioxidants. The above bioactivities could be attributed to the presence of several therapeutically significant class of compounds as revealed by the biochemical profiling and the GC-MS analysis. These constituents, individually or in combination, may account for the pharmacological actions of the extract. Low SI for several compounds in the ADR1 metabolite preparations was noteworthy as it indicated greater possibility of finding novel molecules.

## Figures and Tables

**Figure 1 microorganisms-08-00929-f001:**
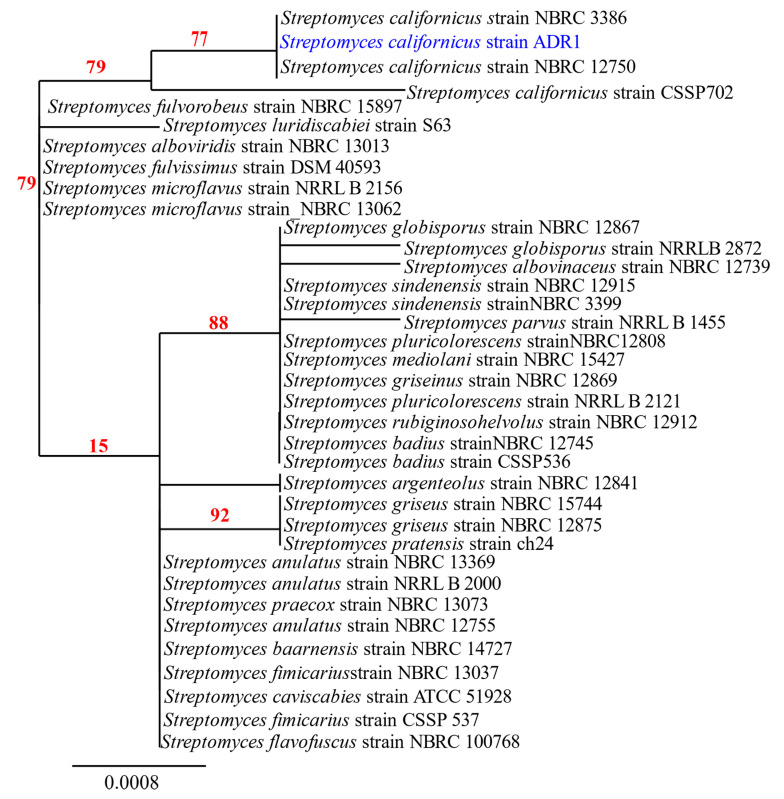
Phylogenetic analysis of isolate ADR1. Neighbour-joining phylogenetic tree showed maximum likelihood model showing the phylogenetic relationship of selected isolate (highlighted in blue) based on 16S rRNA gene sequence alignments. The numbers at the branching points are the percentages of occurrence in 500 bootstrapped trees. Bar indicated 0.0008 substitutions per nucleotide position.

**Figure 2 microorganisms-08-00929-f002:**
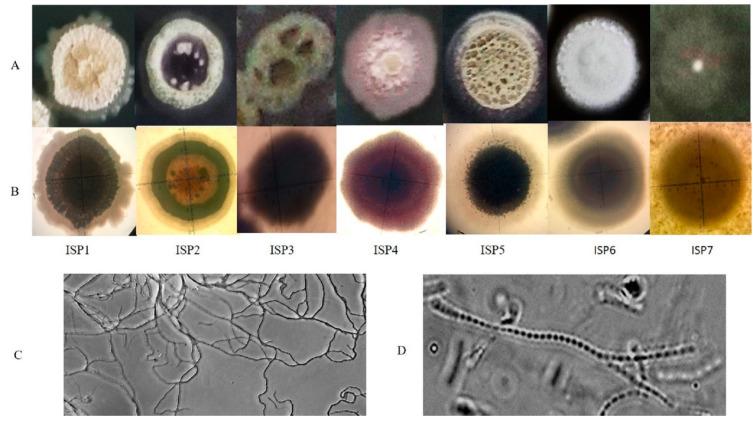
Morphological characterization of *S. californicus* ADR1. Colony observations were made on different ISP media. Panel (**A**) represented the magnified view showing exterior structures of the single colony while panel (**B**) showed interior view of the colony as observed under a Nikon Stereo Zoom Microscope SMZ1270; the mycelial network of branched hyphae could be observed in panel (**C**) and arrangement of spores in chains were viewed in panel (**D**).

**Figure 3 microorganisms-08-00929-f003:**
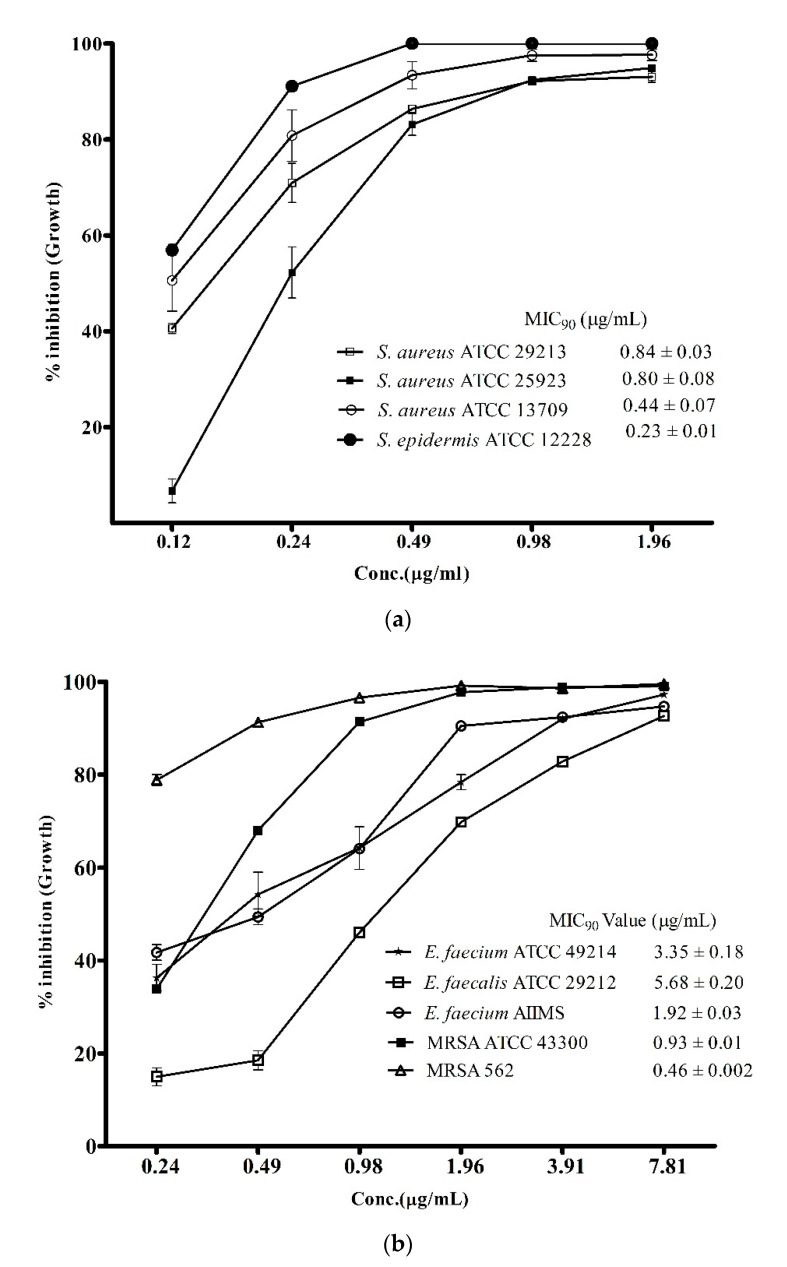
Potency of ADR1 metabolites for Gram-positive pathogens. Minimum inhibitory concentration (MIC_90_) values of the metabolites against the target pathogens were corelated with its antibacterial potency. Percent inhibition in growth at different concentration of the metabolites was measured against the reference strains as shown in (**a**) and in (**b**). The data represented mean ± SEM values of experiments done in triplicate (*p* value < 0.0001).

**Figure 4 microorganisms-08-00929-f004:**
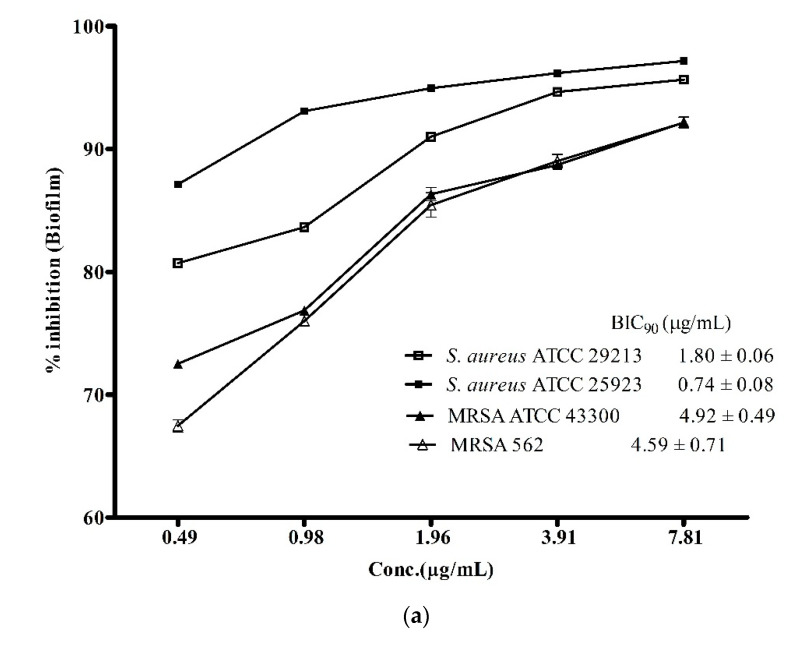
Anti-biofilm activity of ADR1 metabolites. (**a**). Inhibition of biofilm formation and (**b**) inhibition of pre-formed biofilm of *S. aureus* ATCC 29213, 25923, MRSA 562 and ATCC 43300 are shown at various concentration of the ADR1 metabolites. The data represented mean ± SEM values of experiments done in triplicate (*p* value < 0.0001).

**Figure 5 microorganisms-08-00929-f005:**
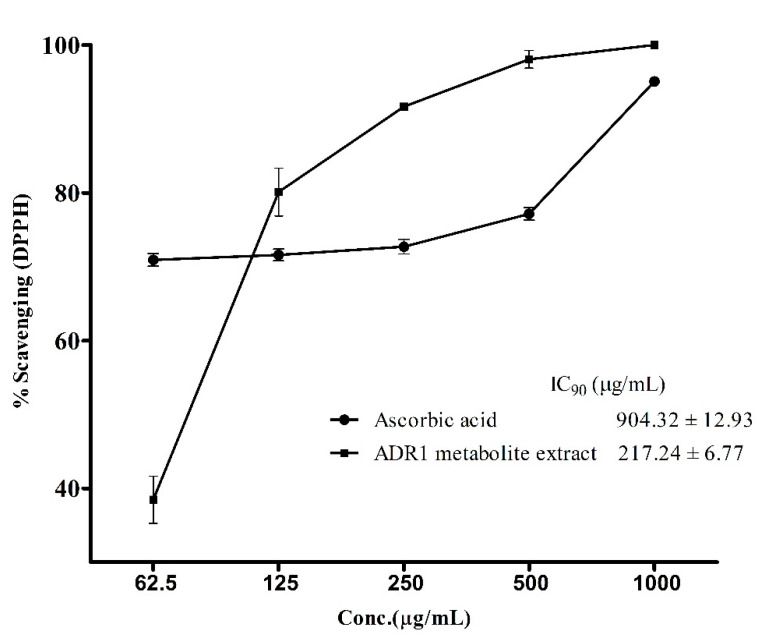
DPPH (1, 1-diphenyl-2-picrylhydrazyl) radical scavenging by ADR1 metabolites. Oxidation of DPPH free radicals were measured at the different concentrations of the metabolite. The data represented mean ± SEM values of the experiments done in triplicate (*p* value < 0.0001).

**Figure 6 microorganisms-08-00929-f006:**
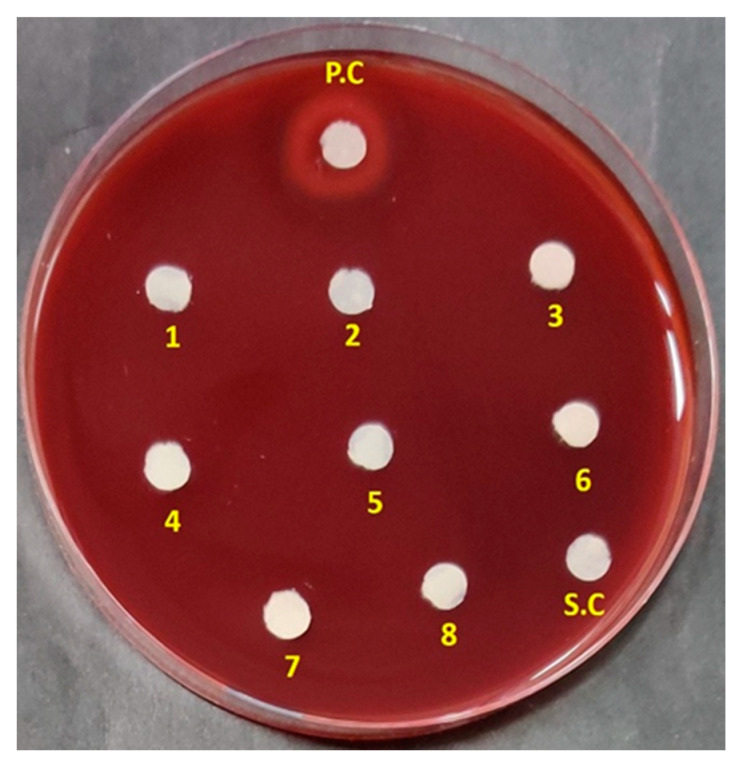
Haemolytic activity of the ADR1 metabolites. Spot No. 1 to 8 showed metabolite dilutions from higher (1000 µg/mL) to lower concentration (7.8125 µg/mL). ‘PC’ was positive control with 0. 1% SDS. SC represented solvent control.

**Table 1 microorganisms-08-00929-t001:** Cultural characteristics of the isolate ADR1 on international *Streptomyces* project (ISP) media.

ISP Media	Growth	Substrate Mycelium	Aerial Mycelium	Diffusible Pigments	Appearance
ISP-1 (Tryptone-Yeast Extract Broth)	Abundant	Crimson red	White	No	Shrinked and depressed with irregular edges
ISP-2 (Yeast extract- Malt extract Agar)	moderate	Wine Red	Pale green	Yellow	Elevated, smooth, regular edges
ISP-3 (Oatmeal agar)	Abundant	Wine Red	Dusty green	Light violet	Shrinked, pits formation, regular edges
ISP-4 (Inorganic salt starch agar)	Moderate	Dark pink	Light pink	Light pink	Flat, wavy edges, pointed centre
ISP-5 (Glycerol asparagine agar base)	Abundant	Pink	Dusty green	No	Elevated, Round, smooth edges
ISP-6 (Peptone yeast extract iron agar)	Moderate	Rusty red	White	Light pink	Elevated at centre, Round, smooth edges
ISP-7 (Tyrosine agar)	Scanty	Light pink	Whitish Pink	No	Pin pointed at centre, flat, round and smooth edges

**Table 2 microorganisms-08-00929-t002:** Spectrum of antibacterial activity of the metabolites produced by *S. californicus* ADR1.

S. No.	Reference Strains of Gram-Positive Pathogens	Zone of Inhibition (mm)
1	*Staphylococcus aureus* ATCC 29213	22.5 ± 0.58
2	*S. aureus* ATCC 25923	19 ± 0.42
3	*S. aureus* ATCC 13709	20 ± 0.5
4	*S. epidermis* ATCC 12228	18 ± 0.45
5	Methicillin-resistant *S. aureus* (MRSA) ATCC 43300	21.3 ± 0.27
6	MRSA 562 (clinical strain)	19 ± 0.25
7	*Enterococcus faecium* ATCC 49224	16.5 ± 0.4
8	*E. faecium* AIIMS	19.4 ± 0.47
9	*E. faecalis* ATCC 29212	17 ± 0.52

**Table 3 microorganisms-08-00929-t003:** Chemical profiling of ethyl acetate extract of secondary metabolites produced by *S. californicus* strain ADP4.

Chemical Class of Metabolites	Testes/Reagents Used	Observations	Results
Terpenoids	Salkowski Test	Reddish brown coloration at the interface	+
Phenols	Folin–Ciocalteu Test	Blue coloration was appeared	+
Flavonoids	Ferric chloride Test	Formation of greenish colour	+
	NaOH, HCl	Intense yellow coloration after adding HCl	
Terpenes	Salkowski Regent	Appearance of golden colour in the chloroform layer	+
Alkaloids	Wagner’s Test	Formation of reddish-brown precipitate	+
Anthocyanins	HCl, Ammonia	Appearance of pink-red, turns blue	+
Anthraquinones	H_2_SO_4_, Chloroform, Ammonia	Light pink coloured layer of ammonia	+
Glycosides	Keller–Killiani Test	A reddish-brown colour ring at the junction of the two layers	+
Tannins	Lead Acetate Test	No precipitation was observed	−
Saponins	Foam Test	No frothing was observed	−
Lactones	Pyridine, sodium nitroprusside, NaOH	No change in coloration was observed	−
Coumarins	Alcoholic NaOH	Yellow fluorescence was not appeared on the paper soaked in NaOH	−
Sterols	Salkowski Test	No red colour was appeared in the lower layer (two layers formed)	−
Lignins	Gallic acid	No appearance of Olive-green coloration	−
Carbohydrates	Fehling’s Test	No reddish violet ring appeared	−
Fatty acids	Ether	No appearance of transparence on filter paper	−
Proteins	Biuret Test	No violet coloration was observed	−

Note: ‘+’: Present; ‘−’: Absent.

**Table 4 microorganisms-08-00929-t004:** Similarity Index-based analysis of the compounds in the secondary metabolite extract of *S. californicus* ADR1. The data were obtained from GC-MS chromatogram. The reference compounds were from NIST and Wiley Library.

S. No.	RT (min)	Similarity Index (SI)	Reference Compounds	Chemical class	Therapeutic Properties
1	11.945	88	Methanoazulen-9-ol, decahydro-2,2,4,8-tetramethyl-stereoisomer	Sesquiterpene (alpha-Caryophyllene alcohol)	Antibacterial, antioxidant, antiinflammatory [73]
2	13.089	96	Naphtho[2,3-g]-1,6,2,5-dioxasilaborocin	Organoboranic acid	No activity reported
3	13.231	85	2-[(trimethylsilyl)oxy]-4-methoxyacetophenone	Flavonoid	No activity reported
4	14.519	90	5-z-methyl-2-z-hydroxycarbonyl-5-e-ethenyl-4-z-propen-2-ylcyclohexanone	Terpenoids (Asperaculane B type)	Anticonvulsant drug design pharmacophore, sodium channel blocker, GABA-transaminase inhibitors [76]
5	15.102	92	1,2-benzenedicarboxylic acid	Diisobutyl phthalate	No activity reported
6	16.068	88	2-pyridineethanamine, n-methyl-n-[2-(4-pyridinyl)ethyl]	Alkaloid (betahistine types)	Vasodilation and reduction of endolymphatic pressure [79]

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
