# Peer review of "Isolation and Characterization of a New Endophytic Actinobacterium Streptomyces californicus Strain ADR1 as a Promising Source of Anti-Bacterial, Anti-Biofilm and Antioxidant Metabolites"

_microorganisms, 2020, doi:10.3390/microorganisms8060929_

Round 1
Reviewer 1 Report
The manuscript by Singh and Dubey reports the characterization of an endophytic isolate from Datura metes. This characterization includes the morphology of the best selected strain in several media, the isolation of an active extract, its qualitative and quantitave antibiotic activity, its capacity to inhibit biofilm formation, to produce antioxidants. Further more the authors check the putative toxicity of hemolytic activity. And finally they study the secondary metabolites produced by chemical profiling and GC-MS analysis.
The paper is well written and conceptually developed. The introduction is brief but enough for the content of the manuscript. The materials and methods are well described and the results and discussion are also well done and performed. The figures and tables are clear and the bibliography adequate.
Only some minor comments:
- In Figure 1 the strain in study should be resalted in some way (different colour, bold) and indicated in the legend, Besides, Streptomyces pratensis is not in italics
- line 247 and 516 Streptomyces should be italics
- page 14, lines 413-415 and Tables 4 and Table S1: in the text the corresponding metabolite of RT 13,089 is the one corresponding to 14,380 of the both tables. There is no coincidence among them.
Therefore, I recommend this manuscript for publication
Author Response
Point-wise response to the observations of Reviewer 1:
Only some minor comments:
- In Figure 1 the strain in study should be resalted in some way (different colour, bold) and indicated in the legend, Besides, Streptomyces pratensis is not in italics:
Response: As per the reviewer’s suggestion, the strain in the study is now shown in blue colour in the revised Figure 1 and has also been described in the legend. Name of the organism has been italicized as suggested in the revised figure.
- line 247 and 516 Streptomyces should be italics
Response: The lines corresponding to 247 and 516 are now 246 and 525 respectively in the revised manuscript. Suggested corrections have been incorporated as shown in highlight and track change.
- page 14, lines 413-415 and Tables 4 and Table S1: in the text the corresponding metabolite of RT 13,089 is the one corresponding to 14,380 of the both tables. There is no coincidence among them.
Response: RT of the metabolites have been corrected in the text (page 14, line 413 in revised version) and at point 4 of Table 4 according to the peak appeared in chromatogram (Figure S1). Changes can be seen in the highlights and track changes.
Reviewer 2 Report
The authors presented the bioactivities of the secondary metabolites from Streptmyces californicus strain ADR1. However, this manuscript is not recommended for publication in its present form, but may be reconsidered as a new paper after throughout revisions.
- The introduction part does not describe the purpose of this study. Also, since the content is redundant, describe it briefly.
- The title and content of the manuscript do not match. The novelty of this study is the isolation and identification of Streptomyces californics strain ADR1 from the plant Datura metes. That should be stated in the title.
- The results of the various bioactivity tests conducted in this study are preliminary. It is necessary to isolate each bioactive compound from the extracts used in these tests and determine its chemical structure. The authors consider the constituents from a simple qualitative reaction and GC-MS, but these results alone are not sufficient.
- In Table 3, the authors report sterols, carbohydrates, fatty acids and proteins as "absent". This result is unacceptable as these constituents are considered essential for microorganisms. The authors need to prove that these constituents are not really present.
Author Response
Point-wise response to the observations of Reviewer 2:
- The introduction part does not describe the purpose of this study. Also, since the content is redundant, describe it briefly.
Response: Introduction has been re-written and shown in blue colour in the revised manuscript. The first two paragraphs highlighted the scientific challenges and the background that led to the present study. The third paragraph provided the rational for choosing the experimental organism/system for the present work followed by highlights of the study in the last paragraph.
On account of re-writing of the introduction part, there has been changes in the references. Accordingly, the citations in the text and the reference section have been updated.
- The title and content of the manuscript do not match. The novelty of this study is the isolation and identification of Streptomyces californics strain ADR1 from the plant Datura metes. That should be stated in the title.
Response: The title has been modified as suggested by the reviewer. The modified title has been highlighted in yellow in the revised manuscript.
The title now clearly reflects the content of the manuscript wherein isolation, identification and characterization of the new endophyte has been described followed by characterization of the metabolites produced by strain ADR1 (recovered as ethyl acetate extract).
- The results of the various bioactivity tests conducted in this study are preliminary. It is necessary to isolate each bioactive compound from the extracts used in these tests and determine its chemical structure. The authors consider the constituents from a simple qualitative reaction and GC-MS, but these results alone are not sufficient.
Response: (A). In the present study, metabolite extract was used, which contained several secondary metabolites produced by the isolate ADR1. The bioactivity assays reported in the manuscript are in accordance with the current international scientific practices that are widely reported in recent publications. A few references are cited below as examples for ready reference.
(i). Sharma, P.; Thakur, D. Antimicrobial biosynthetic potential and diversity of culturable soil actinobacteria from forest ecosystems of Northeast India. Sci Rep 2020, 10: 4104.
(ii). Ebani, V.V.; Bertelloni, F.; Najar, B.; Nardoni,S.; Pistelli, L.; Mancianti, F. Antimicrobial activity of essential oils against Staphylococcus and Malassezia strains isolated from canine dermatitis. Microorganisms, 2020, 2: 252.
(iii). Siddharth, S.; Vittal, R.R.; Wink, J.; Steinert, M. Diversity and bioactive potential of actinobacteria from unexplored regions of Western Ghats, India. Microorganisms 2020, 7: 225.
(iv). Ohadi, M.; Forootanfar, H.; Dehghannoudeh, G.; Eslaminejad, T;, Ameri, A.; Shakibaie, M.; Adeli-Sardou, M. Antimicrobial, anti-biofilm, and anti-proliferative activities of lipopeptide biosurfactant produced by Acinetobacter junii B6. Microb Pathog 2020, 138: 103806.
(v). Gacem, M.A.; Ould‐El‐Hadj‐Khelil, A.; Boudjemaa, B.; Wink, J. Antimicrobial and antioxidant effects of a forest actinobacterium V002 as new producer of pectinabilin, undecylprodigiosin and metacycloprodigiosin. Curr Microbiol 2020, Advance online publication.10.1007/s00284-020-02007-1.
(vi) Kemung, H.M.; Tan, L.T.M.; Chan, K.G.; Ser, H.L.; Law, J.W.F.; Lee, L.H.; Goh, B.H. Antioxidant activities of Streptomyces sp. strain MUSC 14 from mangrove forest soil in Malaysia. BioMed Res Int 2020, 6402607: 1-11.
(B). The work on isolation and identification of individual metabolites produced by the culture is presently in progress with the objectives to isolate and to identify the bioactive metabolites produced by the endophyte; and their subsequent characterization for potential therapeutic applications. This part of the study, which is yet to be completed, will be voluminous and would go beyond the scope of the present manuscript.
- In Table 3, the authors report sterols, carbohydrates, fatty acids and proteins as "absent". This result is unacceptable as these constituents are considered essential for microorganisms. The authors need to prove that these constituents are not really present.
Response: The experimental results only suggested the presence or absence of specific class of compounds which could be recovered in the ethyl acetate from the production medium. Ethyl acetate is a solvent of intermediate polarity. Therefore, highly polar compounds, such as proteins and carbohydrates or highly non-polar compounds, such as cholesterols and sterols may not get extracted in ethyl acetate resulting in negative test for these compounds. Further, the extraction protocol is optimized for recovery of those compounds which possess anti-microbial activities and does not aim at recovery of all the compounds synthesized by the organism. The interpretation of the results of the specific chemical tests is limited only to the compounds recovered in the ethyl acetate and not a general conclusion that carbohydrates, proteins, lipids and sterols are not produced by the organism. The clearly laid down objective of this experiment was achieved in a satisfactory manner.
In the light of above discussion, the title of Table 3 has been modified suitably (Line No. 388) to reflect the scope of analysis and interpretation.
Round 2
Reviewer 2 Report
This revised manuscript has been modified according to the reviewer's comments. It is acceptable for publication.